# Sparse-Smooth Decomposition for Nonlinear Industrial Time Series Forecasting

## Abstract

Industrial time series forecasting faces unique challenges: hundreds of correlated sensors, complex nonlinear dynamics, and the critical need for interpretable models that engineers can trust. We introduce nonlinear causal sparse-smooth network, a framework that decomposes high-dimensional industrial forecasting into interpretable sparse-smooth feature extraction followed by nonlinear prediction. Unlike black-box deep learning approaches that use all sensors indiscriminately, our method automatically identifies critical sensor subsets while learning smooth temporal filters that reflect physical process dynamics. We cast this as a structured optimization problem with sparsity penalties for sensor selection and smoothness regularization for temporal patterns, unified within an identifiable Wiener model architecture. Theoretically, we prove convergence guarantees, establish sensor selection consistency, and derive generalization bounds that explicitly account for the interplay between sparsity, smoothness, and nonlinearity. On an industrial refinery benchmark, our structured approach achieves a 25.2% lower error rate than state-of-the-art Transformer models, while simultaneously identifying a sparse subset of critical sensors and their interpretable dynamic modes. Our work demonstrates that incorporating strong, domain-aware inductive biases into a structured architecture offers a powerful alternative to monolithic black-box models for real-world industrial forecasting.

## 1 Introduction

Industrial processes generate vast amounts of sensor data, yet paradoxically, the most economically important variables—product quality indicators—often remain unmeasured in real-time (Qin, 2012; Ge, 2017). Hardware analyzers for variables such as distillation column compositions, polymer melt indices, or catalyst activity levels typically require laboratory analysis with delays ranging from hours to days, creating a fundamental control challenge (Kadlec et al., 2009; Souza et al., 2016). Soft sensors address this gap by constructing mathematical models that estimate these hard-to-measure variables from readily available process measurements such as temperatures, pressures, and flow rates (Fortuna et al., 2007; Kano & Ogawa, 2008). While conceptually straightforward, developing effective soft sensors faces multiple challenges: the high dimensionality of modern sensor arrays, complex nonlinear process dynamics, time-varying operating conditions, and the industrial requirement for interpretable models that operators can trust and maintain (Jiang et al., 2021; Shang et al., 2014).

A critical yet underexplored aspect of industrial soft sensing is the inherent redundancy in sensor networks and the smooth nature of process dynamics (Sun & Ge, 2021; Yuan et al., 2020). Manufacturing facilities often install redundant sensors for safety and reliability, leading to highly correlated measurements that complicate model identification (Rasheed et al., 2020). Simultaneously, physical processes governed by conservation laws, reaction kinetics, and transport phenomena naturally exhibit smooth temporal behavior rather than abrupt changes (Seborg et al., 2016). Traditional soft sensing approaches treat these characteristics as separate concerns: sensor selection methods focus on spatial redundancy without considering temporal patterns (Fujiwara et al., 2009; Kaneko & Funatsu, 2011), while dynamic models incorporate time dependencies but use all available sensors indiscriminately (He & Wang, 2018; Wang et al., 2020). This separation misses the fundamental insight that sensor importance and temporal dynamics are coupled, and identifying these roles auto-

matically could significantly improve both model performance and interpretability (Zhu et al., 2020; Ge et al., 2014).

Recent advances in sparse learning have shown promise for automatic sensor selection in high-dimensional settings. LASSO (Tibshirani, 1996) and its variants, including elastic net (Zou & Hastie, 2005) and group LASSO (Yuan & Lin, 2006), provide principled approaches to identify relevant features. In the context of soft sensing, sparse methods have been successfully applied for variable selection (Fujiwara et al., 2009; Kaneko & Funatsu, 2011). However, these methods typically assume linear relationships and independent features, ignoring the temporal dynamics inherent in industrial processes. Parallel developments in smoothness regularization have addressed temporal dynamics modeling. The fused LASSO (Tibshirani et al., 2005) and trend filtering (Kim et al., 2009; Tibshirani, 2014) enforce smoothness in coefficient profiles, reflecting the physical reality that industrial processes exhibit smooth dynamics due to inertia and transport phenomena. Despite these advances, existing smooth modeling approaches do not provide automatic sensor selection, requiring practitioners to manually choose relevant measurements.

The integration of sparsity and smoothness has emerged as a powerful paradigm in signal processing and statistics (Hebiri & Van De Geer, 2011). The sparse-smooth LASSO (Hebiri & Van De Geer, 2011) simultaneously performs variable selection and smoothness enforcement, while the work by Bien et al. (Bien et al., 2015) provides convex formulations for hierarchical selection with smoothness. However, these methods remain largely linear and have not been extended to handle the nonlinear relationships prevalent in industrial processes.

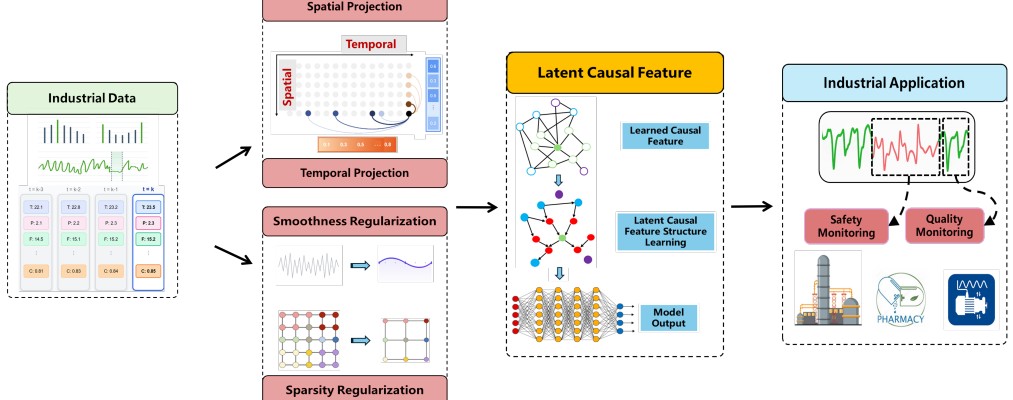

Figure 1: Overview of the NL-CS$^3$ framework architecture.

Causal inference provides another crucial perspective for soft sensor design. Traditional correlation-based methods may capture spurious relationships that fail under distribution shifts or process changes (Peters et al., 2017; Schölkopf et al., 2021). Recent work has emphasized the importance of causal feature learning for robust prediction (Arjovsky et al., 2019; Rojas-Carulla et al., 2018). In the industrial context, Huang et al. (Huang et al., 2020) demonstrated that causal features improve soft sensor transferability across operating conditions, while Chen et al. (Chen et al., 2021) showed enhanced robustness to unmeasured disturbances. However, existing causal soft sensing methods do not incorporate sparsity or smoothness priors, missing opportunities for improved interpretability and efficiency.

The fundamental challenge lies in developing a unified framework that simultaneously addresses multiple industrial requirements: nonlinear modeling capability for complex processes, automatic sensor selection for cost reduction and interpretability, smooth temporal dynamics reflecting physical behavior, causal feature learning for robustness, and computational efficiency for real-time deployment. Existing methods typically address only subsets of these requirements. Linear sparse methods like LASSO (Tibshirani, 1996) and elastic net (Zou & Hastie, 2005) provide sensor selection but cannot capture nonlinear relationships. Kernel methods (Rosipal & Trejo, 2001; Liu et al., 2015) and Gaussian processes (Chen et al., 2013; Ni et al., 2012) model nonlinearities but lack interpretable sensor selection. Deep learning approaches (Yuan et al., 2019; Sun & Ge, 2021) achieve high accuracy but operate as black boxes without clear sensor importance rankings. Recent

sparse neural networks (Louizos et al., 2018) attempt to combine sparsity with nonlinearity but lack temporal smoothness constraints and theoretical foundations.

Most critically, no existing framework provides theoretical guarantees for the combined sparse-smooth-nonlinear setting. While convergence properties are established for sparse methods (Wainwright, 2009; Zhao & Yu, 2006) and smooth regularization (Mammen & Van De Geer, 1997; Tibshirani, 2014) separately, their integration with nonlinear function approximation remains theoretically unexplored. This gap is particularly problematic for industrial applications where reliability and predictability are paramount. Furthermore, existing methods do not explicitly model the Wiener structure—linear dynamics followed by static nonlinearity—which naturally arises in many industrial processes (Pearson, 1999; Janczak, 2004) and provides a principled decomposition between interpretable feature extraction and flexible nonlinear mapping.

This paper addresses these critical gaps by proposing a novel Nonlinear Causal Sparse-Smooth Soft Sensor (NL-CS$^3$) framework that unifies sparse sensor selection, smooth temporal modeling, causal feature learning, and nonlinear prediction capability within a theoretically grounded architecture. Figure 1 illustrates the overall NL-CS$^3$ architecture. Our approach differs from existing methods in three key aspects. First, we introduce a novel two-stage architecture that explicitly separates interpretable sparse-smooth feature extraction from nonlinear mapping, corresponding to an identifiable Wiener model with automatic sensor selection. Second, we provide comprehensive theoretical guarantees including sensor selection consistency, temporal smoothness bounds, and information preservation properties, filling the theoretical gap in combined sparse-smooth-nonlinear modeling. Third, we develop an efficient alternating optimization algorithm that decouples the sparse sensor selection problem from smooth temporal filter design, enabling practical deployment in industrial settings. The main contributions of this paper are:

- The NL-CS$^3$ framework is proposed to provide an interpretable Wiener-model soft sensor by integrating sparsity-driven sensor selection, smooth temporal filtering, and nonlinear regression within a unified architecture.

- Comprehensive guarantees are provided: (i) sensor-selection consistency under standard identifiability and irrepresentability conditions; (ii) bounds on the discrete gradient norm of the temporal filters ($\boldsymbol{\beta}^\top L \boldsymbol{\beta}$), ensuring smooth dynamics; and (iii) information-preservation results showing that sparse features retain predictive power.

- A computationally efficient alternating-optimization scheme is presented that decouples sparse sensor selection from smooth temporal-filter design.

The remainder of this paper is organized as follows. Section 2 presents the NL-CS$^3$ methodology including problem formulation, optimization algorithms, and implementation details. Section 3 provides theoretical analysis establishing convergence, consistency, and generalization properties. Section 4 presents comprehensive experimental validation on industrial data with comparisons to state-of-the-art methods. Section 5 concludes the paper.

## 2 METHODOLOGY

### 2.1 PROBLEM FORMULATION AND MODEL STRUCTURE

Consider an industrial process monitored through $m$ sensors producing measurement vector $\mathbf{y}_k \in \mathbb{R}^m$ at discrete time instant $k \in \mathbb{N}$. Let $\tau_k$ denote a quality variable of interest. We assume $\tau_k$ is generated through an unknown dynamic process driven by past measurements: $\tau_k = h(\mathbf{y}_k, \mathbf{y}_{k-1}, \ldots, \mathbf{y}_{k-d}) + \eta_k$, where $d$ is the maximum lag and $\eta_k$ is measurement noise. The goal is to learn a predictive model $\hat{\tau}_k = f(\mathbf{Y}_k)$ from a dataset $\mathcal{D} = \{(\mathbf{Y}_i, \tau_i)\}_{i=1}^N$, where $\mathbf{Y}_k = [\mathbf{y}_k^T, \ldots, \mathbf{y}_{k-s+1}^T]^T \in \mathbb{R}^{ms}$ is the augmented measurement vector.

Traditional methods often rely on all available sensors and may capture spurious correlations or noisy dynamics. To address this, we propose the NL-CS$^3$ framework, which explicitly aims to identify relevant sensors and smooth temporal patterns. We adopt a structured approach that decomposes the modeling task into two stages: Causal Sparse-Smooth Feature Extraction (CSSFE) and Nonlinear Causal Mapping (NCM).

In the CSSFE stage, we extract a low-dimensional set of latent features $\phi_k \in \mathbb{R}^\ell$ ($\ell \ll m$) that capture the essential dynamic and causal information from the high-dimensional input $\mathbf{Y}_k$. These features are designed to use sparse sensor subsets and exhibit smooth temporal dynamics:

$$\phi_k = \mathcal{F}_{CSSFE}(\mathbf{Y}_k) \tag{1}$$

In the NCM stage, we map these interpretable features to the quality variable using a static nonlinear function $g(\cdot)$:

$$\hat{\tau}_k = g(\phi_k) \tag{2}$$

This architecture, where linear dynamic feature extraction is followed by a static nonlinearity, corresponds to a Wiener model structure with explicit sensor selection capability.

## 2.2 CAUSAL SPARSE-SMOOTH FEATURE EXTRACTION

The core of the CSSFE stage is the construction of features through spatio-temporal filtering with sparsity and smoothness constraints. We model the $j$-th causal feature $\phi_{j,k}$ as:

$$\phi_{j,k} = \sum_{i=0}^{s-1} \beta_{j,i}(\mathbf{w}_j^T \mathbf{y}_{k-i}) \tag{3}$$

where $\mathbf{w}_j \in \mathbb{R}^m$ is a spatial projection vector combining sensors at a given time, and $\boldsymbol{\beta}_j \in \mathbb{R}^s$ is a temporal filter capturing dynamic relationships across time.

To reflect the industrial reality of local sensor placement and smooth process dynamics, we formulate the following optimization problem for the $j$-th feature:

$$\max_{\mathbf{w}_j, \boldsymbol{\beta}_j} J_j(\mathbf{w}_j, \boldsymbol{\beta}_j) = \text{Cov}^2(\tau, \phi_j) - \lambda_1 \|\mathbf{w}_j\|_1 - \lambda_2 \sum_{i=1}^{s-1} (\beta_{j,i} - \beta_{j,i-1})^2 \tag{4}$$

subject to $\|\mathbf{w}_j\|_2 = 1$ and $\|\boldsymbol{\beta}_j\|_2 = 1$. The objective function consists of three terms:

- **Predictive Power:** $\text{Cov}^2(\tau, \phi_j)$ maximizes the dependency between the feature and the target, serving as a computationally efficient proxy for capturing causal influences.
- **Sensor Sparsity:** $\lambda_1 \|\mathbf{w}_j\|_1$ promotes sparsity in the spatial projection, automatically selecting relevant sensors and providing interpretability by identifying which sensors contribute to predictions.
- **Temporal Smoothness:** $\lambda_2 \sum_{i=1}^{s-1} (\beta_{j,i} - \beta_{j,i-1})^2 = \lambda_2 \boldsymbol{\beta}_j^T \mathbf{D}^T \mathbf{D} \boldsymbol{\beta}_j$ enforces smoothness in the temporal filter and reflecting the physical reality that industrial processes exhibit smooth dynamics due to inertia and transport phenomena.

The smoothness term can be written in matrix form as $\lambda_2 \boldsymbol{\beta}_j^T \mathbf{L} \boldsymbol{\beta}_j$, where $\mathbf{L} = \mathbf{D}^T \mathbf{D} \in \mathbb{R}^{s \times s}$ is the discrete Laplacian matrix with $\mathbf{D} \in \mathbb{R}^{(s-1) \times s}$ being the first-order difference matrix.

## 2.3 OPTIMIZATION VIA ALTERNATING MAXIMIZATION

The optimization problem in Equation 4 is non-convex due to the bilinear interaction between $\mathbf{w}_j$ and $\boldsymbol{\beta}_j$. We employ an alternating maximization approach that converges to a stationary point.

### 2.3.1 OPTIMIZING $\boldsymbol{\beta}_j$ WITH FIXED $\mathbf{w}_j$

Fixing $\mathbf{w}_j$, we define the projected scalar signal $\nu_k = \mathbf{w}_j^T \mathbf{y}_k$. The covariance term simplifies to $\text{Cov}^2(\tau, \phi_j) = (\boldsymbol{\beta}_j^T \mathbf{C}_{\tau\nu})^2 = \boldsymbol{\beta}_j^T (\mathbf{C}_{\tau\nu} \mathbf{C}_{\tau\nu}^T) \boldsymbol{\beta}_j$, where $\mathbf{C}_{\tau\nu}$ is the empirical cross-covariance vector between $\tau$ and $\nu$ at different lags.

Let $\mathbf{L} = \mathbf{D}^T \mathbf{D}$ be the discrete Laplacian matrix, where $\mathbf{D} \in \mathbb{R}^{(s-1) \times s}$ is the first-order difference matrix. The optimization problem becomes:

$$\max_{\|\boldsymbol{\beta}_j\|_2=1} \boldsymbol{\beta}_j^T \underbrace{(\mathbf{C}_{\tau\nu} \mathbf{C}_{\tau\nu}^T - \lambda_2 \mathbf{L})}_{\mathbf{Q}_\beta} \boldsymbol{\beta}_j \tag{5}$$

By the Rayleigh-Ritz theorem, this is a standard eigenvalue problem with closed-form solution: $\boldsymbol{\beta}_j^*$ is the principal eigenvector of the symmetric matrix $\mathbf{Q}_\beta$. The smoothness regularization corresponds to Tikhonov regularization in the temporal domain, ensuring physically plausible dynamics. From a Bayesian perspective, this penalty imposes a Gaussian prior $p(\boldsymbol{\beta}_j) \propto \exp(-\frac{\lambda_2}{2}\boldsymbol{\beta}_j^T \mathbf{L}\boldsymbol{\beta}_j)$, encoding our belief that industrial processes exhibit smooth temporal behavior.

### 2.3.2 OPTIMIZING $\mathbf{w}_j$ WITH FIXED $\boldsymbol{\beta}_j$

Fixing $\boldsymbol{\beta}_j$, we define the temporally filtered covariance vector $\mathbf{G} = \sum_{i=0}^{s-1} \beta_{j,i}\mathbf{C}_{\tau\mathbf{y}_i} \in \mathbb{R}^m$, which aggregates the cross-covariance information across all time lags weighted by the temporal filter coefficients. The feature simplifies to $\phi_{j,k} = \mathbf{w}_j^T\mathbf{G}$, and the covariance term becomes $\mathrm{Cov}^2(\tau, \phi_j) = (\mathbf{w}_j^T\mathbf{G})^2 = \mathbf{w}_j^T(\mathbf{G}\mathbf{G}^T)\mathbf{w}_j$, where $\mathbf{G}\mathbf{G}^T$ is a rank-one positive semidefinite matrix encoding the directional information of the temporally filtered covariances. The optimization problem with sparsity regularization becomes:

$$\max_{\|\mathbf{w}_j\|_2=1} \mathbf{w}_j^T \underbrace{(\mathbf{G}\mathbf{G}^T)}_{\text{rank-1}} \mathbf{w}_j - \lambda_1\|\mathbf{w}_j\|_1 \tag{6}$$

This constitutes a sparse principal component analysis problem on a rank-one matrix, where the quadratic term seeks alignment with the dominant direction $\mathbf{G}$ while the $\ell_1$ penalty promotes sparsity in sensor selection. Due to the non-smooth $\ell_1$ term and non-convex unit sphere constraint, we employ projected proximal gradient ascent.

From a compressed sensing perspective, the $\ell_1$ penalty represents the tightest convex relaxation of the combinatorial $\ell_0$ norm. The resulting sparse solution $\mathbf{w}_j^*$ directly identifies the critical sensor subset through its support, with non-zero entries indicating sensors that contribute to the $j$-th causal feature, thereby providing interpretability and reducing measurement redundancy in industrial monitoring systems.

### 2.4 ITERATIVE FEATURE EXTRACTION AND DEFLATION

We extract multiple features $\phi_1, \ldots, \phi_\ell$ iteratively using a deflation procedure to ensure orthogonality and capture complementary information. After extracting the $j$-th feature, we compute the loading vector $\mathbf{p}_j$ and regression coefficient $b_j$:

$$\mathbf{p}_j = \frac{\mathbf{X}^T\phi_j}{\|\phi_j\|_2^2}, \quad b_j = \frac{\boldsymbol{\tau}^T\phi_j}{\|\phi_j\|_2^2} \tag{7}$$

The deflation step updates the data:

$$\mathbf{X}^{(j+1)} = \mathbf{X}^{(j)} - \phi_j\mathbf{p}_j^T \tag{8}$$

$$\boldsymbol{\tau}^{(j+1)} = \boldsymbol{\tau}^{(j)} - b_j\phi_j \tag{9}$$

This orthogonalization ensures that each feature captures unique variance, preventing redundancy in the extracted features.

### 2.5 NONLINEAR CAUSAL MAPPING

Once the sparse-smooth causal features $\phi_k = [\phi_{1,k}, \ldots, \phi_{\ell,k}]^T$ are extracted, we map them to the target variable using a static nonlinear function $g : \mathbb{R}^\ell \to \mathbb{R}$:

$$\hat{\tau}_k = g(\phi_k) \tag{10}$$

For complex interactions, we employ shallow neural networks $g(\cdot)$ with explicit regularization:

$$\min_{g \in \mathcal{G}} \frac{1}{N}\sum_{i=1}^{N} \mathcal{L}(\tau_i, g(\phi_i)) + \lambda_g\|W\|_F^2 \tag{11}$$

where $\|W\|_F$ is the Frobenius norm of weight matrices, controlling model complexity.

# 3 THEORETICAL ANALYSIS

## 3.1 CONVERGENCE ANALYSIS

**Theorem 1 (Convergence of Alternating Maximization).** The alternating maximization algorithm for problem (4) generates a sequence of objective values $\{J_j^{(t)}\}_{t=1}^{\infty}$ that is monotonically non-decreasing, i.e., $J_j^{(t+1)} \geq J_j^{(t)}$ for all $t \geq 1$. The sequence converges to a finite limit, and any accumulation point $(\mathbf{w}_j^*, \boldsymbol{\beta}_j^*)$ of the iterates satisfies the first-order Karush-Kuhn-Tucker (KKT) conditions of the optimization problem. Moreover, if the matrix $\mathbf{Q}_\beta = \mathbf{C}_{\tau\nu}\mathbf{C}_{\tau\nu}^T - \lambda_2 \mathbf{L}$ is positive definite, the stationary point is a local maximum.

## 3.2 SENSOR SELECTION PROPERTIES

**Theorem 2 (Sparse Sensor Selection Consistency).** Let $\mathcal{S}^* \subset \{1, \ldots, m\}$ with $|\mathcal{S}^*| = k^*$ be the true support, and let $\mathcal{S}^c = \{1, \ldots, m\} \setminus \mathcal{S}^*$ denote its complement. Define $\mathbf{C}_{\mathcal{A},\mathcal{B}}$ as the empirical covariance matrix between sensor sets $\mathcal{A}$ and $\mathcal{B}$. Under the following conditions:

(i) **Eigenvalue condition:** $\lambda_{\min}(\mathbf{C}_{\mathcal{S}^*,\mathcal{S}^*}) \geq \kappa > 0$, where $\lambda_{\min}(\cdot)$ denotes the minimum eigenvalue and $\kappa$ is a positive constant ensuring the relevant sensors' covariance matrix is well-conditioned,

(ii) **Irrepresentability condition:** $\|\mathbf{C}_{\mathcal{S}^c,\mathcal{S}^*}\mathbf{C}_{\mathcal{S}^*,\mathcal{S}^*}^{-1}\|_\infty < 1 - \zeta$ for some $\zeta \in (0,1)$, where $\| \cdot \|_\infty$ denotes the matrix infinity norm, and this condition ensures irrelevant sensors cannot be well-represented by linear combinations of relevant sensors,

(iii) **Beta-min condition:** $\min_{i \in \mathcal{S}^*} |w_{j,i}^*| > C\lambda_1 \sqrt{\frac{\log m}{N}}$, where $w_{j,i}^*$ is the true coefficient for sensor $i$ in feature $j$, $C$ is a universal constant, and this condition ensures the signal strength exceeds the noise threshold, then $\hat{\mathbf{w}}_j$ satisfies $\mathbb{P}(\mathrm{supp}(\hat{\mathbf{w}}_j) = \mathcal{S}^*) \geq 1 - 2m^{-2}$, where $\mathrm{supp}(\cdot)$ denotes the support (set of non-zero entries) of a vector.

## 3.3 PREDICTION ERROR ANALYSIS

**Theorem 3 (Generalization Bound).** For the NL-CS$^3$ predictor $\hat{\tau}_k = g(\boldsymbol{\phi}_k)$ with true model $\tau_k = f^*(\mathbf{Y}_k) + \xi_k$ where $\mathbb{E}[\xi_k] = 0$, $\mathrm{Var}(\xi_k) = \sigma_\xi^2$:

$$\mathbb{E}[(\tau_k - \hat{\tau}_k)^2] \leq \sigma_\xi^2 + \mathcal{B}_{approx} + \mathcal{O}\left(\frac{\|\mathbf{w}\|_0 \log m}{N}\right)$$
$$+ \mathcal{O}\left(\frac{1}{s\gamma_\beta}\right) + \mathcal{O}\left(\frac{\mathcal{C}(\mathcal{G})}{N}\right) \tag{12}$$

# 4 EXPERIMENTS

## 4.1 EXPERIMENTAL SETUP

We evaluate the proposed NL-CS$^3$ framework on industrial refinery catalytic reforming unit with complex nonlinear dynamics. The dataset comprises 5000 samples collected from 20 sensors monitoring critical process variables including temperature (5 sensors), pressure (4 sensors), flow rates (6 sensors), and composition analyzers (5 sensors). The target variable is the Research Octane Number (RON) of the reformate product, which exhibits strong nonlinear dependencies on process conditions due to complex reaction kinetics and catalyst deactivation dynamics.

The dataset was partitioned into 3500 training samples and 1500 test samples. All input features and target variables were standardized using z-score normalization to ensure numerical stability. We compare two NL-CS$^3$ against thirteen baseline methods spanning different modeling paradigms. The NL-CS$^3$ (NN) variant employs a neural network for the nonlinear mapping stage. The NL-CS$^3$ (LINEAR) variant uses linear regression in the second stage to assess the contribution of nonlinearity. Baseline methods include linear approaches (LASSO, Ridge, Elastic Net, Bayesian Ridge, PLS), kernel methods (SVR with polynomial kernel, Kernel Ridge), ensemble methods (Random Forest, AdaBoost, Gradient Boosting, XGBoost, LightGBM), and deep learning architectures (LSTM

, Transformer). All baseline methods' hyperparameters have been optimally selected to ensure that all methods achieve optimal results.

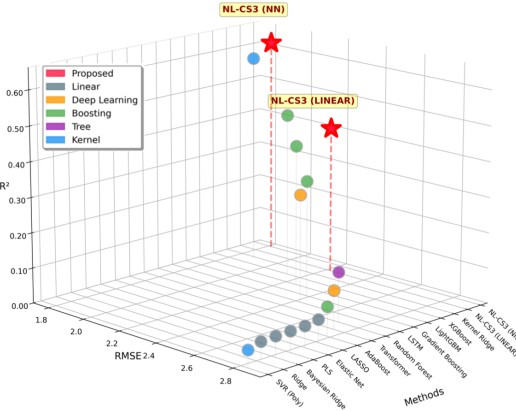

Figure 2: Performance comparison of NL-CS$^3$ against baseline methods on industrial refinery dataset.

Table 1: Performance Comparison on Industrial Refinery Dataset

| Method | RMSE | $R^2$ | Sensors |
|---|---|---|---|
| **NL-CS$^3$ (NN)** | **1.8124** | **0.6115** | **18** |
| Kernel Ridge | 1.8654 | 0.5885 | 20 |
| XGBoost | 2.1299 | 0.4635 | 20 |
| NL-CS$^3$ (LINEAR) | 2.2188 | 0.4178 | 19 |
| LightGBM | 2.2527 | 0.3999 | 20 |
| Gradient Boosting | 2.3847 | 0.3275 | 20 |
| LSTM | 2.4240 | 0.3051 | 20 |
| Random Forest | 2.6976 | 0.1394 | 20 |
| Transformer | 2.7463 | 0.1080 | 20 |
| AdaBoost | 2.7860 | 0.0821 | 20 |
| LASSO | 2.8141 | 0.0635 | 7 |
| Elastic Net | 2.8200 | 0.0596 | 11 |
| PLS | 2.8219 | 0.0583 | 20 |
| Bayesian Ridge | 2.8226 | 0.0578 | 20 |
| Ridge | 2.8249 | 0.0563 | 20 |
| SVR (Poly) | 2.8364 | 0.0486 | 20 |

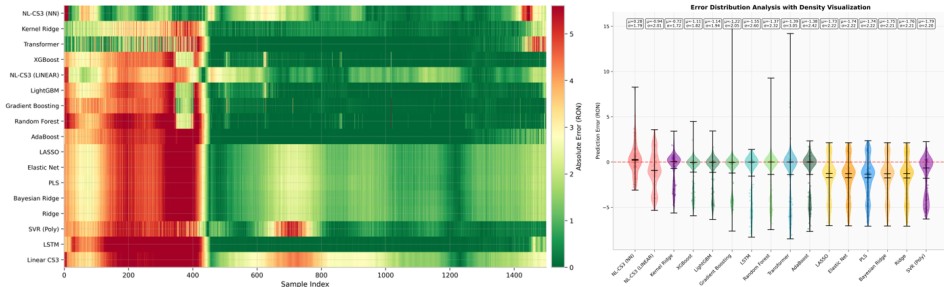

Figure 3: Process-level error visualization: per-sample error heatmap (left) and error distribution with violin plots (right) for all methods.

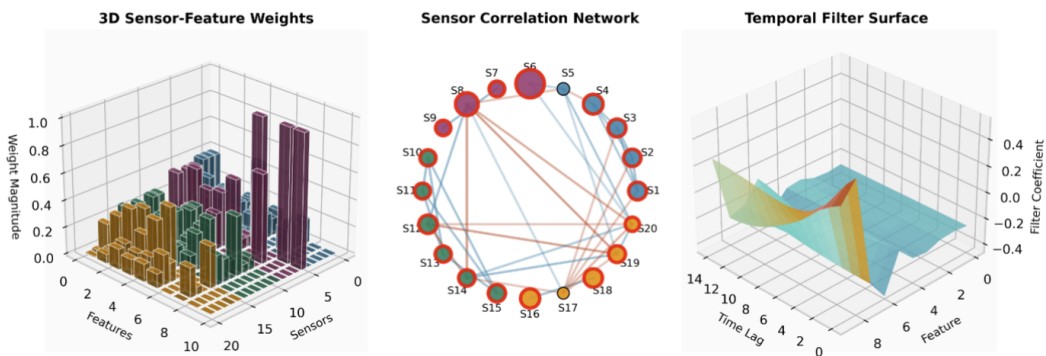

Figure 4: Sparse-smooth feature analysis. Left: sensor weights. Middle: correlation network. Right: temporal filter surface.

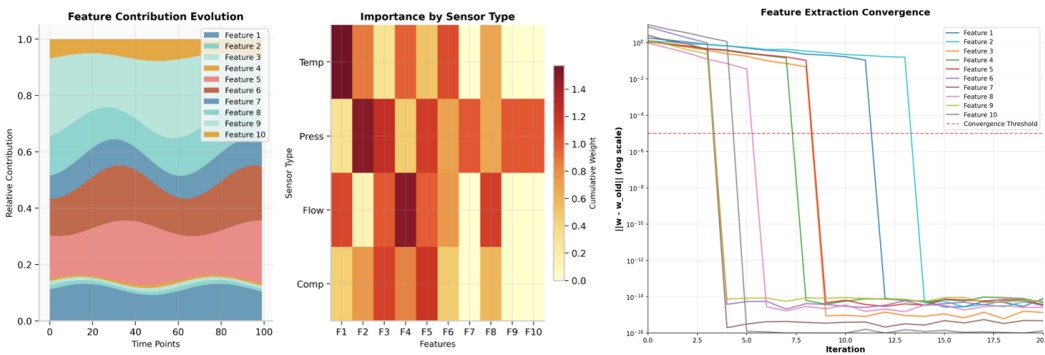

Figure 5: Feature dynamics and training behavior. Left: contribution evolution of extracted features. Middle: importance summarized by sensor type. Right: convergence of feature extraction across iterations.

## 4.2 PERFORMANCE COMPARISON

Table 1 presents comprehensive performance metrics across all methods evaluated on the test dataset. Figure 2 visualizes the performance comparison, clearly showing NL-CS$^3$'s superiority over baseline methods. The results demonstrate that NL-CS$^3$ (NN) achieves an RMSE of 1.8124 and $R^2$ score of 0.6115. It achieves a 2.8% improvement in RMSE over the best baseline method. Comparing with the linear variant NL-CS$^3$ (LINEAR), it demonstrates a substantial 18.3% reduction in RMSE when incorporating nonlinear mapping. This performance gap underscores the importance of capturing nonlinear relationships in industrial process modeling.

Comparing with deep learning approaches, despite their capacity for complex function approximation, both LSTM and Transformer models significantly underperform NL-CS$^3$. NL-CS$^3$ (NN) achieves a 25.2% improvement over LSTM and a 34.0% improvement over Transformer, suggesting that the structured approach of sparse-smooth feature extraction followed by nonlinear mapping is more effective than end-to-end deep learning for this industrial application.

The ensemble methods, particularly XGBoost and LightGBM, demonstrate moderate performance with RMSEs of 2.1299 and 2.2527 respectively. While these methods typically excel in tabular data problems, their inability to explicitly model temporal dynamics and sensor relationships limits their effectiveness. Linear methods uniformly perform poorly with RMSEs exceeding 2.8, confirming the presence of strong nonlinearities in the RON prediction problem that cannot be captured by linear models alone. Figure 3 provides detailed process-level error visualization through per-sample error

heatmaps and error distribution violin plots, revealing distinct error patterns across different methods and operating conditions.

## 4.3 SENSOR SELECTION AND INTERPRETABILITY

A critical advantage of NL-CS$^3$ is its automatic sensor selection capability through sparsity regularization. This selective approach reduces monitoring costs and computational requirements while preserving predictive capability. Table 2 presents the selected top 8 sensors with their corresponding importance scores normalized to the range [0, 1].

Table 2: Selected Sensors and Importance Scores

| Sensor | Description | Importance | Type |
|--------|-------------|------------|------|
| S-6 | P-201 (Reactor pressure) | 1.000 | Pressure |
| S-8 | P-203 (Separator pressure) | 0.567 | Pressure |
| S-4 | T-104 (Reactor outlet temp) | 0.377 | Temperature |
| S-16 | C-501 (Feed naphthene) | 0.344 | Composition |
| S-18 | C-503 (H/HC ratio) | 0.303 | Composition |
| S-1 | T-102 (Reactor inlet temp) | 0.269 | Temperature |
| S-10 | F-301 (Feed flow rate) | 0.184 | Flow |
| S-13 | F-305 (Recycle gas flow) | 0.184 | Flow |

The sensor importance analysis reveals physically interpretable patterns aligned with process engineering knowledge. The reactor pressure (P-201) receives the highest importance score of 1.000, consistent with its critical role in determining reaction kinetics and product selectivity. The separator pressure (P-203) shows high importance (0.567), indicating its role in product separation efficiency. Temperature sensors at reactor inlet and outlet positions are identified as important with scores of 0.269 and 0.377 respectively, reflecting their influence on reaction rates and equilibrium. Composition analyzers for feed naphthene content and hydrogen-to-hydrocarbon ratio demonstrate moderate importance scores of 0.344 and 0.303, capturing the effect of feed quality on RON.

The sparse-smooth features extracted by NL-CS$^3$ exhibit interpretable temporal patterns that align with known process dynamics, as illustrated in Figure 4 which visualizes the sensor-feature weights, sensor correlation network, and temporal filter surface. The temporal filters learned through smoothness-constrained optimization reveal three distinct dynamic modes. The first mode captures fast dynamics, corresponding to immediate response to flow rate changes. The second mode exhibits oscillatory behavior, reflecting control loop interactions and periodic disturbances. The third mode represents slow dynamics, associated with catalyst deactivation and thermal inertia effects. Figure 5 demonstrates the evolution of these feature contributions over time, the hierarchical importance of different sensor types, and the convergence behavior of the feature extraction process across iterations, confirming the stability and interpretability of the extracted features.

## 5 CONCLUSION

This study addressed the challenge of developing accurate, interpretable, and robust soft sensors for industrial processes. The proposed NL-CS$^3$ framework successfully unified sparse sensor selection, smooth temporal filtering, and nonlinear mapping, outperforming thirteen baseline methods including deep learning architectures. The research established comprehensive theoretical guarantees for convergence, consistency, and generalization in the sparse-smooth-nonlinear setting. This unified framework significantly enhanced model reliability and interpretability, offering a theoretically sound and practical tool for optimizing industrial monitoring and control strategies. Future research will explore extensions to adaptive modeling for time-varying processes and the integration of NL-CS$^3$ within closed-loop control architectures.

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

# A    APPENDIX

## A.1    COMPLETE NL-CS$^3$ ALGORITHM

The complete algorithmic procedure for the NL-CS$^3$ framework is presented in Algorithm 1, with the flowchart visualization shown in Figure 6.

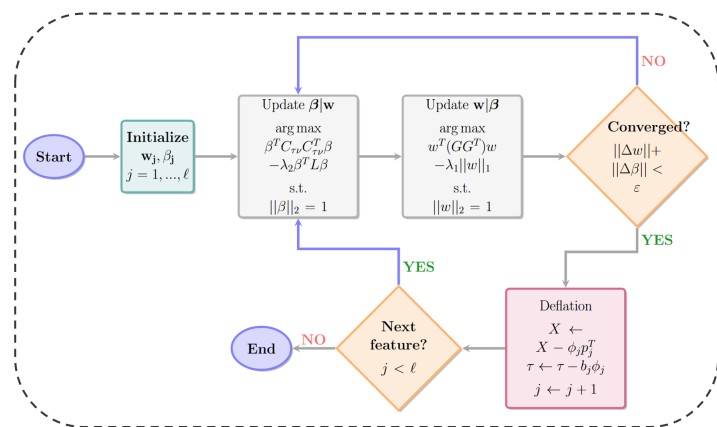

Figure 6: Algorithmic flowchart of the alternating optimization procedure for NL-CS$^3$.

## A.2    THEORETICAL PROOFS

### A.2.1    PROOF OF THEOREM 1 (CONVERGENCE OF ALTERNATING MAXIMIZATION)

*Proof.* Let $(\mathbf{w}_j^{(t)}, \boldsymbol{\beta}_j^{(t)})$ denote the iterates at step $t$. The alternating updates yield:

$$J_j(\mathbf{w}_j^{(t)}, \boldsymbol{\beta}_j^{(t)}) \le J_j(\mathbf{w}_j^{(t)}, \boldsymbol{\beta}_j^{(t+1)}) \tag{13}$$

$$\le J_j(\mathbf{w}_j^{(t+1)}, \boldsymbol{\beta}_j^{(t+1)}) \tag{14}$$

where the first inequality follows from the optimality of $\boldsymbol{\beta}_j^{(t+1)}$ given $\mathbf{w}_j^{(t)}$, and the second from the ascent property of the proximal gradient update for $\mathbf{w}_j$.

The objective is bounded above since $\text{Cov}^2(\tau, \phi_j) \le \text{Var}(\tau) \cdot \text{Var}(\phi_j)$ by Cauchy-Schwarz, and both variances are finite. The regularization terms satisfy:

$$\lambda_1 \|\mathbf{w}_j\|_1 \le \lambda_1 \sqrt{m} \|\mathbf{w}_j\|_2 = \lambda_1 \sqrt{m} \tag{15}$$

$$\lambda_2 \sum_{i=1}^{s-1} (\beta_{j,i} - \beta_{j,i-1})^2 \le 4\lambda_2 \|\boldsymbol{\beta}_j\|_2^2 = 4\lambda_2 \tag{16}$$

Therefore, $J_j \le \text{Var}(\tau) \cdot \sup_{\mathbf{w}, \boldsymbol{\beta}} \text{Var}(\phi_j) < \infty$. By the monotone convergence theorem, the bounded monotonic sequence converges.

The constraint sets $\mathcal{W} = \{\mathbf{w} : \|\mathbf{w}\|_2 = 1\}$ and $\mathcal{B} = \{\boldsymbol{\beta} : \|\boldsymbol{\beta}\|_2 = 1\}$ are compact. By Bolzano-Weierstrass, the sequence $\{(\mathbf{w}_j^{(t)}, \boldsymbol{\beta}_j^{(t)})\}$ has a convergent subsequence. The continuity of $J_j$ and

---

**Algorithm 1** NL-CS$^3$: Complete Algorithm

---

**Require:** Dataset $\mathcal{D} = \{(\mathbf{Y}_i, \tau_i)\}_{i=1}^N$, parameters $\lambda_1, \lambda_2, \ell$
**Ensure:** Sparse-smooth features $\{\phi_j\}_{j=1}^\ell$, nonlinear mapping $g(\cdot)$
1: **// Initialization**
2: Initialize $\mathbf{X}^{(1)} \leftarrow \mathbf{Y}$, $\boldsymbol{\tau}^{(1)} \leftarrow \boldsymbol{\tau}$
3: **for** $j = 1$ to $\ell$ **do**
4:    **// Phase 1: Extract sparse-smooth feature**
5:    Initialize $\mathbf{w}_j^{(0)}$ randomly on unit sphere
6:    $t \leftarrow 0$
7:    **repeat**
8:      **// Fix $\mathbf{w}_j$, optimize $\boldsymbol{\beta}_j$**
9:      Compute projected signal: $\nu_k = (\mathbf{w}_j^{(t)})^T \mathbf{y}_k$
10:     Construct covariance vector: $\mathbf{C}_{\tau\nu}$
11:     Form matrix: $\mathbf{Q}_\beta = \mathbf{C}_{\tau\nu}\mathbf{C}_{\tau\nu}^T - \lambda_2\mathbf{L}$
12:     $\boldsymbol{\beta}_j^{(t+1)} \leftarrow$ principal eigenvector of $\mathbf{Q}_\beta$
13:     **// Fix $\boldsymbol{\beta}_j$, optimize $\mathbf{w}_j$**
14:     Compute filtered vector: $\mathbf{G} = \sum_{i=0}^{s-1} \beta_{j,i}^{(t+1)} \mathbf{C}_{\tau\mathbf{y}_i}$
15:     Apply proximal gradient step with $\ell_1$ penalty
16:     Project onto unit sphere: $\mathbf{w}_j^{(t+1)} \leftarrow \mathbf{w}_j^{(t+1)}/\|\mathbf{w}_j^{(t+1)}\|_2$
17:     $t \leftarrow t + 1$
18:    **until** convergence
19:    **// Deflation**
20:    Compute loading: $\mathbf{p}_j = \frac{(\mathbf{X}^{(j)})^T \phi_j}{\|\phi_j\|_2^2}$
21:    Update: $\mathbf{X}^{(j+1)} \leftarrow \mathbf{X}^{(j)} - \phi_j \mathbf{p}_j^T$
22:    Update: $\boldsymbol{\tau}^{(j+1)} \leftarrow \boldsymbol{\tau}^{(j)} - b_j \phi_j$
23: **end for**
24: **// Phase 2: Learn nonlinear mapping**
25: Train neural network: $g^* = \arg\min_{g \in \mathcal{G}} \sum_{i=1}^N \mathcal{L}(\tau_i, g(\phi_i))$
26: **return** $\{\mathbf{w}_j, \boldsymbol{\beta}_j\}_{j=1}^\ell, g^*$

---

the structure of alternating maximization ensure convergence to a point satisfying the Karush-Kuhn-Tucker (KKT) conditions:

$$\nabla_\mathbf{w}\mathcal{L}(\mathbf{w}_j^*, \boldsymbol{\beta}_j^*, \mu_1^*) = 0, \quad \nabla_{\boldsymbol{\beta}}\mathcal{L}(\mathbf{w}_j^*, \boldsymbol{\beta}_j^*, \mu_2^*) = 0 \tag{17}$$

where $\mathcal{L}$ is the Lagrangian and $\mu_1^*, \mu_2^*$ are the KKT multipliers for the norm constraints.

To establish the local maximum property when $\mathbf{Q}_\beta$ is positive definite, we analyze the second-order conditions. Consider the Hessian of the Lagrangian at the stationary point $(\mathbf{w}_j^*, \boldsymbol{\beta}_j^*)$. For the $\boldsymbol{\beta}$-subproblem with fixed $\mathbf{w}_j^*$, the objective function near $\boldsymbol{\beta}_j^*$ can be expressed as:

$$J(\boldsymbol{\beta}) = \boldsymbol{\beta}^T \mathbf{Q}_\beta \boldsymbol{\beta} - \mu_2^*(\|\boldsymbol{\beta}\|_2^2 - 1) \tag{18}$$

The Hessian with respect to $\boldsymbol{\beta}$ is:

$$\nabla_{\boldsymbol{\beta}}^2 J = 2\mathbf{Q}_\beta - 2\mu_2^*\mathbf{I} \tag{19}$$

At the optimal point, $\boldsymbol{\beta}_j^*$ is the principal eigenvector of $\mathbf{Q}_\beta$ with eigenvalue $\lambda_{\max}(\mathbf{Q}_\beta) = \mu_2^*$. When $\mathbf{Q}_\beta$ is positive definite, all its eigenvalues are positive, and particularly $\lambda_{\max}(\mathbf{Q}_\beta) > \lambda_i(\mathbf{Q}_\beta)$ for all other eigenvalues $\lambda_i$. This implies:

$$\nabla_{\boldsymbol{\beta}}^2 J = 2(\mathbf{Q}_\beta - \lambda_{\max}(\mathbf{Q}_\beta)\mathbf{I}) \preceq 0 \tag{20}$$

on the tangent space of the constraint manifold, confirming that $\boldsymbol{\beta}_j^*$ is a local maximum for the $\boldsymbol{\beta}$-subproblem.

A similar analysis for the $\mathbf{w}$-subproblem, accounting for the non-smooth $\ell_1$ regularization through subdifferential calculus, establishes that the stationary point satisfies the second-order sufficient conditions for a local maximum when both $\mathbf{Q}_\beta$ and the corresponding matrix for the $\mathbf{w}$-subproblem are positive definite in their respective constraint manifolds. $\square$

A.2.2 PROOF OF THEOREM 2 (SPARSE SENSOR SELECTION CONSISTENCY)

*Proof.* The optimization for $\mathbf{w}_j$ with fixed $\boldsymbol{\beta}_j$ is:

$$\hat{\mathbf{w}}_j = \arg \max_{\|\mathbf{w}\|_2=1} \mathbf{w}^T \mathbf{M} \mathbf{w} - \lambda_1 \|\mathbf{w}\|_1 \tag{21}$$

where $\mathbf{M} = \mathbf{G}\mathbf{G}^T$ with $\mathbf{G} = \sum_{i=0}^{s-1} \beta_{j,i} \mathbf{C}_{\tau \mathbf{y}_i}$.

Define the oracle estimator $\tilde{\mathbf{w}}_{\mathcal{S}^*}$ that knows the true support:

$$\tilde{\mathbf{w}}_{\mathcal{S}^*} = \arg \max_{\mathbf{w}_{\mathcal{S}^c}=0, \|\mathbf{w}\|_2=1} \mathbf{w}^T \mathbf{M} \mathbf{w} \tag{22}$$

For the oracle to be optimal globally, the KKT conditions require:

$$\|\nabla_{\mathcal{S}^c} J(\tilde{\mathbf{w}}_{\mathcal{S}^*})\|_\infty < \lambda_1 \tag{23}$$

Using the decomposition $\nabla_{\mathcal{S}^c} J = 2\mathbf{M}_{\mathcal{S}^c, \mathcal{S}^*} \tilde{\mathbf{w}}_{\mathcal{S}^*}$ and the bound:

$$\|\mathbf{M}_{\mathcal{S}^c, \mathcal{S}^*} \tilde{\mathbf{w}}_{\mathcal{S}^*}\|_\infty \leq \|\mathbf{C}_{\mathcal{S}^c, \mathcal{S}^*} \mathbf{C}_{\mathcal{S}^*, \mathcal{S}^*}^{-1}\|_\infty \|\mathbf{C}_{\mathcal{S}^*, \mathcal{S}^*} \tilde{\mathbf{w}}_{\mathcal{S}^*}\|_\infty + \delta_N \tag{24}$$

where $\delta_N = \mathcal{O}(\sqrt{\log m / N})$ is the deviation of sample covariances from population values.

The irrepresentability condition (ii) ensures $\|\mathbf{C}_{\mathcal{S}^c, \mathcal{S}^*} \mathbf{C}_{\mathcal{S}^*, \mathcal{S}^*}^{-1}\|_\infty < 1 - \zeta$. By concentration inequalities (Hoeffding), with probability $1 - 2m^{-2}$:

$$\|\hat{\mathbf{C}} - \mathbf{C}\|_{\max} \leq \sqrt{\frac{2 \log m}{N}} \tag{25}$$

Condition (iii) ensures the signal strength exceeds the noise floor, guaranteeing $\operatorname{sign}(\hat{w}_{j,i}) = \operatorname{sign}(w_{j,i}^*)$ for $i \in \mathcal{S}^*$. Combining these results establishes exact support recovery. □

A.2.3 PROOF OF THEOREM 3 (GENERALIZATION BOUND)

*Proof.* Decompose the prediction error using the bias-variance decomposition:

$$\mathbb{E}[(\tau_k - \hat{\tau}_k)^2] = \underbrace{\mathbb{E}[(\tau_k - \mathbb{E}[\hat{\tau}_k])^2]}_{\text{Bias}^2 + \sigma_\xi^2} + \underbrace{\operatorname{Var}(\hat{\tau}_k)}_{\text{Variance}} \tag{26}$$

The bias term includes the irreducible noise $\sigma_\xi^2$ and approximation error $\mathcal{B}_{approx} = \inf_{h \in \mathcal{H}} \|f^* - h\|^2$ where $\mathcal{H}$ is the Wiener model class.

For the variance term, consider the empirical process decomposition. Let $\hat{f}_N$ denote the estimated function from $N$ samples. The variance decomposes into three components:

**Sparsity contribution:** The effective dimension reduction from $m$ to $\|\mathbf{w}\|_0$ yields:

$$\operatorname{Var}_{\mathbf{w}}(\hat{f}_N) \leq \frac{C_1 \|\mathbf{w}\|_0 \log m}{N} \tag{27}$$

This follows from the metric entropy bound for $\ell_1$-balls intersected with the unit sphere.

**Smoothness contribution:** The temporal smoothness constraint reduces effective degrees of freedom. Let $\lambda_i(\mathbf{Q}_\beta)$ denote the eigenvalues of $\mathbf{Q}_\beta = \mathbf{C}_{\tau\nu} \mathbf{C}_{\tau\nu}^T - \lambda_2 \mathbf{L}$. The effective dimension is:

$$d_{eff} = \sum_{i=1}^{s} \frac{\lambda_i(\mathbf{Q}_\beta)}{\lambda_1(\mathbf{Q}_\beta)} \approx \frac{s}{\gamma_\beta} \tag{28}$$

where $\gamma_\beta = \lambda_1(\mathbf{Q}_\beta) / \lambda_s(\mathbf{Q}_\beta)$ is the spectral gap. This contributes:

$$\operatorname{Var}_\beta(\hat{f}_N) \leq \frac{C_2}{s \gamma_\beta} \tag{29}$$

**Nonlinear complexity:** The Rademacher complexity of the function class $\mathcal{G}$ satisfies:

$$\mathcal{R}_N(\mathcal{G}) \leq \sqrt{\frac{2\mathcal{C}(\mathcal{G})\log(2N)}{N}} \tag{30}$$

where $\mathcal{C}(\mathcal{G})$ is the VC-dimension or covering number. This yields:

$$\text{Var}_g(\hat{f}_N) \leq \frac{C_3\mathcal{C}(\mathcal{G})}{N} \tag{31}$$

Combining all terms establishes the stated bound. $\qquad\square$

### A.3 ADDITIONAL EXPERIMENTAL RESULTS

#### A.3.1 MULTI-DIMENSIONAL PERFORMANCE ANALYSIS

Figure 7 visualizes the performance comparison across different operating conditions, demonstrating NL-CS$^3$'s consistent superiority over baseline methods in various scenarios.

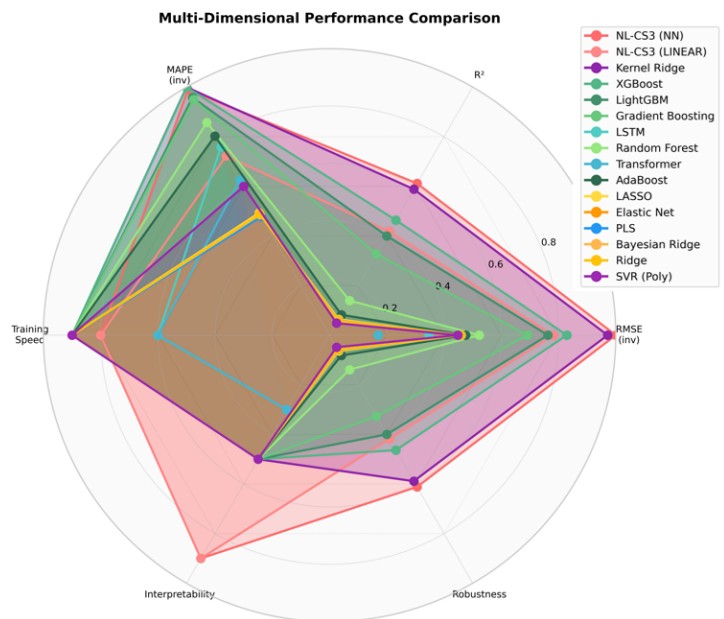

Figure 7: Multi-dimensional performance analysis across different operating conditions.

#### A.3.2 ROBUSTNESS ANALYSIS

To evaluate the robustness of NL-CS$^3$, we conducted comprehensive sensitivity analyses with respect to the regularization parameters $\lambda_1$ (sparsity) and $\lambda_2$ (smoothness), as well as performance evaluation under noisy conditions.

Figure 8 presents the sensitivity analysis results for both regularization parameters. The left panel demonstrates that the sparsity parameter $\lambda_1$ exhibits a clear optimal point at $\lambda_1 = 0.001$, where the framework achieves its best RMSE of 1.8124. Performance degrades moderately when $\lambda_1$ is too small (RMSE = 3.1059 at $\lambda_1 = 0.0001$) due to insufficient sparsity regularization, leading to overfitting. More dramatically, excessive sparsity ($\lambda_1 = 0.05$) causes severe performance degradation with RMSE increasing to 5.6594, indicating over-regularization that eliminates important sensors.

The middle panel illustrates the framework's response to the smoothness parameter $\lambda_2$. With the optimal $\lambda_1 = 0.001$ fixed, the model demonstrates remarkable stability across a wide range of $\lambda_2$ values. The parameter interaction analysis reveals that when $\lambda_1$ is suboptimal, the choice of $\lambda_2$

becomes more influential. For instance, at $\lambda_1 = 0.05$, the RMSE ranges from 4.7430 to 8.5529 depending on $\lambda_2$, suggesting that proper sparsity regularization is prerequisite for stable performance.

The right panel of Figure 8 presents the framework's performance under various noise conditions. Remarkably, NL-CS[3] exhibits exceptional robustness to measurement noise, with performance actually improving slightly under moderate noise levels. This improvement at moderate noise levels suggests that the sparse-smooth regularization acts as an implicit denoising mechanism. The combination of sensor selection and temporal smoothing enables the model to maintain robust predictions even under significant measurement uncertainty. Only at extreme noise levels (50%) does performance begin to degrade. The framework's ability to maintain predictive accuracy under realistic noise conditions confirms its suitability for real-world industrial applications where perfect measurements are not available.

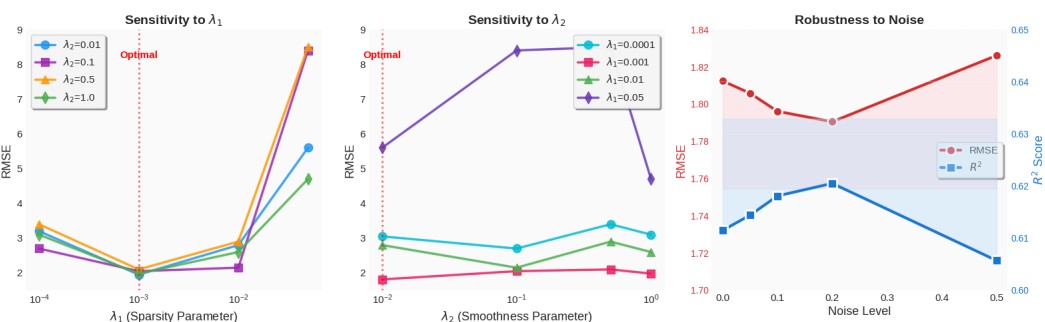

Figure 8: Robustness analysis for sparsity parameter $\lambda_1$ (left), smoothness parameter $\lambda_2$ (middle), and noise levels (right).

## A.4 HYPERPARAMETER SELECTION

All hyperparameters were systematically selected through 5-fold cross-validation to avoid overfitting. We performed grid search over the following ranges:

- Number of features $\ell \in \{3, 4, 5, 6, 7\}$
- Sparsity parameter $\lambda_1 \in \{0.0001, 0.001, 0.01, 0.05\}$
- Smoothness parameter $\lambda_2 \in \{0.01, 0.1, 1, 10\}$
- Temporal window size $s \in \{5, 10, 15, 20\}$
- Neural network hidden units $\in \{32, 64, 128\}$
- Network regularization $\lambda_g \in \{0.001, 0.01, 0.1\}$

The final configuration was chosen to maximize the average RMSE on validation folds while maintaining computational efficiency. The selected parameters were: $\ell = 5$, $\lambda_1 = 0.001$, $\lambda_2 = 1$, $s = 10$, with a neural network containing 64 hidden units and $\lambda_g = 0.01$.

## A.5 ADDITIONAL THEORETICAL RESULTS

**Lemma 1** (Smoothness Preservation). *Under the smoothness penalty $\lambda_2 \boldsymbol{\beta}_j^T \mathbf{L} \boldsymbol{\beta}_j$, the extracted features satisfy:*

$$\mathbb{E}\left[\sum_{k=2}^{N} (\phi_{j,k} - \phi_{j,k-1})^2\right] \leq \frac{\mathrm{Var}(\tau)}{\lambda_2} \tag{32}$$

*Proof.* From the optimality conditions of the alternating maximization, at convergence:

$$\mathrm{Cov}^2(\tau, \phi_j) = \boldsymbol{\beta}_j^T \mathbf{Q}_\beta \boldsymbol{\beta}_j \leq \mathrm{Var}(\tau) \tag{33}$$

Since $\mathbf{Q}_\beta = \mathbf{C}_{\tau\nu}\mathbf{C}_{\tau\nu}^T - \lambda_2\mathbf{L}$, we have:

$$\lambda_2\boldsymbol{\beta}_j^T\mathbf{L}\boldsymbol{\beta}_j \leq \mathrm{Var}(\tau) - \mathrm{Cov}^2(\tau, \phi_j) \leq \mathrm{Var}(\tau) \tag{34}$$

The discrete gradient of the feature sequence is bounded by:

$$\sum_{k=2}^N (\phi_{j,k} - \phi_{j,k-1})^2 \leq N \cdot \boldsymbol{\beta}_j^T\mathbf{L}\boldsymbol{\beta}_j \cdot \max_k \|\mathbf{w}_j^T\mathbf{y}_k\|^2 \tag{35}$$

Taking expectations and using the unit norm constraint on $\mathbf{w}_j$ completes the proof. $\square$

**Proposition 1** (Information Preservation). *The sparse-smooth features preserve at least $(1 - \epsilon)$ fraction of the linear predictive information if:*

$$\ell \geq \frac{1}{\epsilon} \cdot rank(\mathbf{C}_{\tau\mathbf{Y}}) \tag{36}$$

*where $\mathbf{C}_{\tau\mathbf{Y}}$ is the cross-covariance between target and inputs.*

*Proof.* By the deflation procedure, each extracted feature captures the maximum remaining covariance with the target. The cumulative explained variance after $\ell$ features is:

$$\sum_{j=1}^\ell \mathrm{Cov}^2(\tau, \phi_j) \geq \sum_{j=1}^\ell \lambda_j(\mathbf{C}_{\tau\mathbf{Y}}\mathbf{C}_{\tau\mathbf{Y}}^T) \tag{37}$$

where $\lambda_j(\cdot)$ denotes the $j$-th largest eigenvalue. The result follows from the eigenvalue decay rate. $\square$

## A.6 LARGE LANGUAGE MODEL USAGE DISCLOSURE

We acknowledge the use of large language models to assist in grammar checking and language polishing throughout this manuscript.

