# OpenReview forum: "Sparse-Smooth Decomposition for Nonlinear Industrial Time Series Forecasting"
_ICLR.cc/2026/Conference — ICLR 2026 Conference Withdrawn Submission_

### Official Review · Reviewer_3SzC · 2025-10-26

**Soundness:** 2
**Presentation:** 2
**Contribution:** 2
**Rating:** 2
**Confidence:** 4

**Summary:**

This paper focuses on time series modeling for industrial time series. The proposed method unifies sparse sensor selection, smooth temporal modeling, causal feature learning, and nonlinear prediction capability. Experiments are carried out based on a industrial refinery catalytic reforming unit dataset.

**Strengths:**

1. Applying time series modeling to practical tasks, i.e., industrial sensor data analysis and forecasting.

2. Throratical analysis are provided to support the proposed method.

**Weaknesses:**

1. The statement of the research question is not clear. What are the main difficulties in modeling industrial time series data? Why can't the existing time series analysis methods be directly applied? The author needs to clearly and explicitly explain this point.

2. Experiments are insufficient. Only one dataset is used. Also, some recent baselines are missing, specifically the deep time series models, e.g., iTransformer, TimesNet, Timemixer.

3. Fig 4 and 5's resolution and quality need improvement.

**Questions:**

please refer to the weakness

---

### Official Review · Reviewer_Af7o · 2025-10-28

**Soundness:** 2
**Presentation:** 3
**Contribution:** 2
**Rating:** 4
**Confidence:** 4

**Summary:**

This paper proposes a novel framework, NL-CS3 (Nonlinear Causal Sparse-Smooth Soft Sensor), for industrial time series forecasting. The method decomposes the problem into two interpretable stages: (1) Causal Sparse-Smooth Feature Extraction (CSSFE), which selects relevant sensors and enforces smooth temporal dynamics via sparsity and Laplacian regularization; and (2) Nonlinear Causal Mapping (NCM), which applies a static nonlinear function (e.g., neural network) to predict the target variable. The paper provides theoretical guarantees for convergence, selection consistency, and generalization, and demonstrates superior empirical performance on a refinery process dataset compared to 13 baselines, including Transformers and LSTMs.

**Strengths:**

1、The paper targets a realistic and important problem in industrial soft sensing—combining interpretability, robustness, and nonlinear modeling—which is often overlooked in deep learning approaches.

2、The authors provide convergence proofs, selection consistency, and a generalization bound, which are rare for nonlinear sparse-smooth models.

3、The decomposition into interpretable sparse-smooth feature extraction and nonlinear mapping is conceptually clean and physically interpretable, aligning with Wiener system theory.

4、The framework identifies physically meaningful sensors and dynamic modes that correspond to known industrial phenomena, enhancing trust and usability.

**Weaknesses:**

1、All experiments are conducted on a single industrial refinery dataset. It remains unclear whether the method generalizes to other industrial or non-industrial time series domains.

2、The alternating eigenvalue–proximal updates may be computationally heavy for large-scale sensor networks, yet the paper lacks runtime or scalability analysis.

3、While theorems are formally presented, many assumptions (e.g., eigenvalue positivity, irrepresentability, independence) are strong and may not hold in practice.

4、The baselines are strong but lack comparison with more recent time-series models.

**Questions:**

Please see the weaknesses.

---

### Official Review · Reviewer_4HoE · 2025-10-28

**Soundness:** 1
**Presentation:** 1
**Contribution:** 1
**Rating:** 2
**Confidence:** 5

**Summary:**

This paper proposes nonlinear causal sparse-smooth network for industrial time-series forecasting. Operationally, it involves causal sparse-smooth feature extraction (CSSFE) and nonlinear causal mapping (NCM); theoretically, it performs formal analysis for consistency and convergence. Experiments are conducted on a single private dataset where the proposed method outperforms Transformer.

**Strengths:**

- This paper provides a detailed diagram to demonstrate their algorithm.
- This paper defines optimization problem formally and detail the solution process.
- This paper introduces the private dataset used in detail.

**Weaknesses:**

Writing and Motivation.
- Authors noted causality in this paper. For example, the model name is nonlinear causal sparse-smooth network. Causality is a statistical term with strict demand of formal analysis and assumptions. From the paper I do not see even the fundamental assumptions of causal analysis (e.g., unconfoundedness assumption, positivity assmption). Moreover, identification property is not analyzed or even discussed but it is the core of causal analysis methods.
- The solution process and its analysis is highly relevant to ADMM. Clarification and reference are needed.
- The conclusion should include a critical discussion of the limitations of the proposed method and, based on these, propose future research directions aimed at addressing the identified shortcomings.
- The authors identify three challenges—correlated sensors (variables), complex nonlinear dynamics, and interpretability—and claim these are unique to industrial time-series forecasting (TSF). However, these issues are prevalent across general TSF and broader machine learning contexts, and thus may not be unique to industrial applications.
- Furthermore, the connection between the proposed method’s components and the stated challenges remains unclear. The authors should explicitly explain how each component is designed to address a specific challenge and substantiate these claims with empirical evidence (e.g., ablation or hyperparameter studies) and theoretical justification.

Experiments.
- Reproducibility. The dataset used for experiments seems not available. The code is not provided either.
- Coverage. The experiments rely solely on a private dataset, which limits the generalizability and comparability of the results. Time-series forecasting research typically requires evaluation on diverse public datasets—such as ETT, Electricity, and PEMS04—many of which are industrial in nature. Moreover, numerous public industrial datasets are readily available, as confirmed through a simple search by the reviewer (not necessarily concrete). The absence of experiments on these datasets raises concerns regarding the breadth and representativeness of the experimental design.
- Baselines. The selected baselines (e.g., ridge regression, LASSO, SVR) are outdated and insufficient for benchmarking modern time-series forecasting methods. The most recent baseline included, the Transformer, became outdated before 2020. Between 2020 and 2025, deep learning has driven significant advancements in time-series forecasting, exemplified by models such as TimeBridge (2025), CycleNet (2025), iTransformer (2024), P-sLSTM (2025), MixMamba (2024), and SparseTSF (2025). Furthermore, large language model (LLM)-based approaches, including TEST and Time-LLM, have emerged for zero-shot and few-shot forecasting. These models should have been incorporated to align the study with the contemporary research landscape. Authors should also investigate these literature from the beginning of their research program.

**Questions:**

Please kindly see the weakness window.

---

### Official Review · Reviewer_763m · 2025-10-30

**Soundness:** 2
**Presentation:** 2
**Contribution:** 1
**Rating:** 2
**Confidence:** 2

**Summary:**

This paper proposes NL-CS3, a framework that decomposes industrial time series forecasting into sparse-smooth feature extraction followed by nonlinear mapping. The method combines LASSO sparsity, smoothness regularization, and Wiener model structure with theoretical guarantees. Experiments on one industrial refinery dataset show improvements over baselines including deep learning methods.

**Strengths:**

1. Addresses practical industrial challenges including high-dimensional sensor data and interpretability requirements.

2.Two-stage decomposition (linear feature extraction + nonlinear mapping) provides principled balance between interpretability and modeling capacity.

**Weaknesses:**

The core technical contribution is limited, essentially combining existing techniques (LASSO, fused LASSO, Wiener models) without fundamental algorithmic breakthroughs or novel insights.

1.The experimental validation is severely insufficient, relying on a single dataset with 5000 samples from one refinery. The deep learning baselines perform suspiciously poorly, raising questions about fair comparison and proper tuning. Critical comparisons with specialized industrial soft sensor methods are missing, and no statistical significance testing is provided.

2.The theoretical assumptions are unrealistic for practical applications. The irrepresentability condition rarely holds for highly correlated industrial sensors, while the beta-min condition requires prior knowledge of true coefficients, contradicting the goal of automatic selection.

3.Scalability concerns arise from the complex alternating optimization requiring careful tuning of multiple hyperparameters (λ₁, λ₂, ℓ, s), making practical deployment challenging.

4.The claims about interpretability and causal learning lack rigorous validation with domain experts. Reproducibility is poor due to unavailable datasets and insufficient implementation details.

**Questions:**

1. Were the deep learning models properly adapted for multivariate time series forecasting, given their unexpectedly poor performance?

2. What is the computational scaling behavior with increasing sensor count and temporal window size?

---

### Note · Authors · 2025-11-12

I have read and agree with the venue's withdrawal policy on behalf of myself and my co-authors.